# Application of Keratograph and Fourier-Domain Optical Coherence Tomography in Measurements of Tear Meniscus Height

**DOI:** 10.3390/jcm11051343

**Published:** 2022-02-28

**Authors:** Minjie Chen, Anji Wei, Jianjiang Xu, Xingtao Zhou, Jiaxu Hong

**Affiliations:** 1Department of Ophthalmology, Eye and Ear, Nose and Throat Hospital of Fudan University, 83 Fenyang Road, Shanghai 200031, China; minjie.chen@fdeent.org (M.C.); anji.wei@fdeent.org (A.W.); jianjiang.xu@fdeent.org (J.X.); 2Key Laboratory of Visual Impairment and Restoration of Shanghai, Fudan University, 83 Fenyang Road, Shanghai 200031, China; 3Key Myopia Laboratory of NHC, Fudan University, 83 Fenyang Road, Shanghai 200031, China; 4Key Laboratory of Myopia, Chinese Academy of Medical Science, 83 Fenyang Road, Shanghai 200031, China; 5Department of Ophthalmology, Affiliated Hospital of Guizhou Medical University, 28 Guiyi Road, Guiyang 200031, China

**Keywords:** dry eye, tear meniscus, keratograph, optical coherence tomography

## Abstract

To compare the interoperator repeatability of tear meniscus height (TMH) measurements obtained with a keratograph and Fourier-domain optical coherence tomography (FD-OCT) and to assess the agreement between the methods.Forty-seven eyes with DED and 41 healthy eyes were analyzed using the Schirmer test I and tear breakup time test (TBUT). The TMH was measured three times with each device. The repeatability of measurements was assessed by within-subject standard deviation (Sw), repeatability (2.77 Sw), coefficient of variation (CoV) and intraclass correlation coefficient (ICC). Efficacy in detecting DED was evaluated in terms of the area under the curve (AUC). The TMHs obtained with the keratograph were 0.03 mm lower than those obtained with FD-OCT in both groups (*p* < 0.001 for the DED group and *p* = 0.0143 for the control group, respectively). The intraexaminerICCs of the keratographic TMH were 0.789 and 0.817 for the DED and control groups, respectively, and those of the FD-OCT TMH were 0.859 and 0.845, respectively. Although a close correlation was found between the TMHs measured with the keratograph and FD-OCT by the Spearman analysis in both groups (both *p* < 0.001), poor agreement between the devices was shown in both groups using a Bland–Altman plot. The AUCs of the keratography and FD-OCT results were 0.971 (*p* < 0.001) and 0.923 (*p* < 0.001), respectively. Both devices had excellent diagnostic accuracy in differentiating normal patients from DED patients. FD-OCT TMH measurements were more reliable than the keratograph data in the DED group. Agreement between the devices was poor in both groups.

## 1. Introduction

Dry eye disease (DED) is a very common ocular comorbidity with a reported prevalence from 5% to 35% in adults [1]. Most diagnostic tests are aimed at measuring the unstable tear film and decreased tear volume. Although no gold standard exists, the Schirmer test I is the most frequently used method for checking changes in tear volume. However, this conventional test is invasive and is usually influenced by reflex tearing, which has shown poor diagnostic sensitivity and repeatability [2]. It would be ideal to have noninvasive or minimally invasive objective measurements to help standardize the clinical assessment of DED and metrics, which would also provide better outcome measures for monitoring the effects of treatment [3]. The lower eyelid tear meniscus can be measured for tear meniscus height (TMH); this technique has shown relatively high sensitivity and specificity [4,5]. The Dry Eye Workshop II (DEWS II) report suggested that TMH could be used for the diagnosis of aqueous-deficient DED [5,6,7,8]. Several methods are available for quantifying the tear meniscus, including slit-lamp evaluation with a graticule scale, reflective meniscometry and video assessment [5]. Nevertheless, these methods are frequently not clinically available due to the complexity of the procedures and low-accuracy repeatability.

In addition to the methods mentioned above, optical coherence tomography (OCT) and the new keratograph (OculusOptikgerate GmbH, Wetzlar, Germany) have been widely applied in studies of TMH measurements in a noninvasive, noncontact and rapid manner [5,7,9]. The utilization of techniques with infrared light and anterior segment OCT using low coherence light allows the TMH to be measured without reflex tearing, which has improved its accuracy [10,11,12]. In previous studies, significant correlations between TMH by OCT and vital staining scores, Schirmer test I values and tear film breakup time (TBUT) were observed, indicating that OCT is a noninvasive and practical method for the quantitative evaluation of tear fluid which has the potential for detecting dry eye and suspected dry eye [10,11,12]. For another, the noncontact keratograph of two versions can show high-resolution images of the lower meniscus and provide a simple, noninvasive screening test for the TMH and TBUT with acceptable sensitivity, specificity and repeatability [7,13,14]. Although several studies have assessed the agreement of TMH measurement between the OCT and Keratograph 5M, different conclusions have been drawn [5,15]. Data from one previous study came from healthy subjects, whereas another study used DED patients. The different study populations may account for the differences in the results. Consequently, we aimed to determine the intraobserver repeatability and diagnostic efficacy of TMH measurements provided by a keratograph (Keratograph 5M; OCULUS, Inc., Wetzlar, Germany) and a Fourier-domain OCT (FD-OCT; RTVue-100; Optovue, Inc., Freemont, CA, USA) in both healthy and DED subjects and to assess the agreement between the two devices, as well as exploring their possible relationship with traditional dry-eye examinations.

## 2. Materials and Methods

### 2.1. Patients

This is an observational cross-sectional study of 47 Chinese subjects with DED attending a tertiary eye clinic and 41 normal control subjects recruited from a population-based study. The right eye from each subject was chosen as the study eye. A full ophthalmological examination, including uncorrected visual acuity, intraocular pressure and slit-lamp examination, was taken in all subjects. Meanwhile, the TBUT test and Schirmer test I (30 mm; Tianjin Jingming New Technological Development Co., Ltd., Tianjin, China) without topical anesthesia were performed. The diagnosis of DED was made if the subject exhibited all of the following characteristics: significant subjective symptoms graded as 3 or more according to the questionnaire (Table 1) [7] and either a TBUT ≤ 10 s or a Schirmer test I ≤ 5 mm/5 min. Asymptomatic subjects with both a TBUT > 10 s and a Schirmer test I > 5 mm were enrolled in the healthy group. This study was approved by the institutional review board of the Eye and ENT Hospital. All procedures conformed to the tenets of the Declaration of Helsinki. Informed consent was obtained from all subjects.

Subjects with Sjögren’s syndrome, pterygium, severe conjunctivitis or blepharitis, nasolacrimal obstruction, or cornea opacity were excluded. Those with a history of contact lens wearing, ocular surgery, or punctual occlusion were also excluded. Anther exclusion criterium was the use of eye medications or artificial tears during the previous month.

Subjects were examined between 8:00 a.m. and 11:30 a.m. in a room with controlled temperature (26–27 °C) and humidity (30–50%). The FD-OCT and keratograph were placed side-by-side in the same room to shorten the session. Each instrument was operated and measured by a single examiner who was masked to the study. The order in which the images were obtained was randomized. We repeated each measurement three times with both machines. Subsequently, the TBUT and Schirmer test I contact examinations were performed by another examiner. The examination procedures for the TBUT and Schirmer test I were described in our previous study [7]. Each check interval of rest time was between 10 and 15 min.

### 2.2. TMH Measurement with FD-OCT

An FD-OCT system ((FD-OCT; RTVue-100; Optovue, Inc., Freemont, CA, USA) with a corneal adaptor module was used in the current study. Scans were taken exactly below the corneal vertex, centered on the inferior cornea and lower eyelid. The subject was asked to blink normally during both imaging procedures. Vertical images were recorded three times for 3 sec after each blink at the 6 o’clock position of the cornea. Three images were obtained for each patient. A built-in caliper was used to measure the TMH. The TMH was determined as the length from the point where the meniscus intersected with the cornea superiorly to the eyelid inferiorly (Figure 1A).

### 2.3. Keratographic Measurements

The keratograph illuminates with infrared light-emitting diodes to ensure a dark examination environment and avoid reflex tearing. All subjects were instructed to blink normally before the images of the tear meniscus were captured. For each eye, the examination was performed three times with a scanning interval from 3 to 5 sec. The measurement of the lower TMH was performed at the 6 o’clock zone between the cornea and the lower eyelid. The TMH measurement was determined as the length between the darker edge of the lower eyelid and the upper border of the reflex line of the tear meniscus (Figure 1B). The keratograph was loaded with built-in measurement software to improve the accuracy of the TMH measurement.

### 2.4. Statistical Analysis

For the descriptive statistical analysis, we used Excel 2007 with SPSS for Windows, Version 19.0.1 (IBM Corp., Armonk, NY, USA). The results are provided as the mean ± SD. Within-subject standard deviation (Sw), test–retest repeatability (2.77 Sw), coefficient of variation (CoV) and intraclass correlation coefficients (ICCs) were calculated for three repeated measurements by each examiner to determine the intraexaminer repeatability of each device. The CoV was calculated as the ratio of the Sw to the overall mean. The ICCs of both examiners were evaluated according to the Cronbach’s alpha score. If the score was >0.8, the correlation was accepted as reliable. Receiver operating characteristic (ROC) curves with calculations of the area under the curve (AUC) and cut points were used to describe the accuracy of the TMH measurements obtained with both devices. A Bland–Altman analysis was constructed to evaluate the agreement between devices. The Spearman correlation coefficient was used to determine the correlations between variables. The paired t-test or the matched-pair signed-rank test was used to identify between-group differences. Numeration data were compared between the two groups using χ^2^ tests. Significance was set at *p* ≤ 0.05.

## 3. Results

### 3.1. Study Population

The patients in the DED group were much older than the patients in the control group (*p* < 0.001), while sex distribution was comparable between groups (*p* = 0.521). Obviously, significant decreases in THM, TBUT and Schirmer tests were seen in the DED group compared with those of the healthy subjects (all *p* < 0.001; Table 2). The average values of the TMH obtained with FD-OCT were larger than those measured with the keratograph in both groups (Table 2; *p* < 0.001 for the DED group and *p* = 0.0143 for the control group, respectively).

### 3.2. Intraoperator Repeatability

Table 3 shows the mean TMH values, Sw, 2.77 Sw, CoV and ICCs recorded using the keratograph and FD-OCT. Our findings indicate worse intraoperator repeatability for the keratograph than for the FD-OCT measurements for both groups. Comparable repeatability in the TMH was observed with FD-OCT in both the DED and control groups, whereas better repeatability for the keratograph measurements was seen in the control group than in the DED group.

### 3.3. Agreement between Devices

The Bland–Altman plot shows the 95% limits of agreement between the keratograph and FD-OCT in both groups (Figure 2). The mean difference between the measurements was 0.03 mm in both groups. The agreement was poor between devices in both groups (Figure 2). The TMH measured with the keratograph correlated well with the TMH measured with FD-OCT in both groups through the Spearman analysis (both *p* < 0.001; Figure 3).

### 3.4. ROC Analysis and Cut-Points

In the ROC analyses, the AUCs of the TMH using the keratograph and FD-OCT were 0.971 (*p* < 0.001) and 0.923 (*p* < 0.001) (Figure 4), respectively, differentiating the patients with DED from those with normal eyes. The cut-points for the TMH were 0.29 mm and 0.32 mm for the keratograph and FD-OCT, respectively.

### 3.5. Correlations between TMH and TBUT or Schirmer Scores

The results for the TMH measurements using the keratograph or FD-OCT correlated with TBUT or Schirmer values are shown in Table 4. Although the TMH with the keratograph weakly positively correlated with the Schirmer test I score (r = 0.4598, *p* = 0.0011) in the DED group, no other correlation was found between the TMH with the keratograph or FD-OCT and TBUT or Schirmer values (Table 4).

## 4. Discussion

In comparison to the traditional Schirmer test I, both the keratograph and FD-OCT can measure the tear quantity noninvasively. Although both devices showed significant diagnostic accuracy, intraoperator repeatability for TMH obtained using the keratograph was not so satisfactory in the DED group. The agreement between devices with the Bland–Altman plot was poor in both groups.

We showed excellent AUCs of more than 0.9 in both devices, indicating good diagnostic efficacy in distinguishing DED subjects from healthy subjects for both methods. The keratograph demonstrated better performance in the AUC analysis, with a score of 0.971, compared to 0.923 for FD-OCT. Moreover, the present results are much better than the previously reported AUC of 0.784 obtained with a keratograph [16]. The lower Schirmer test I value and more severe degree of DED included in our studies may be responsible for this difference. FD-OCT has recently been reported to have a cutoff value of 0.18 mm in the diagnosis of aqueous-deficient dry eye [17]. However, we did not classify DED and obtained a cutoff value of 0.32 mm here. To date, we tried the cutoff points of 0.29 mm with a keratograph for the first time; more research is needed to verify the data.

In the present study, a significant difference of 0.03 mm in the TMH was observed between the measurements obtained with the keratograph and those obtained with FD-OCT in both groups. The TMH measured with the keratograph tended to be lower than the TMH measured with FD-OCT, which is in line with previous studies [5,15]. In published data, the TMH measured with the keratograph was 0.07 mm lower than the TMH measured with FD-OCT in DED patients [15], whereas the difference was 0.01 mm in healthy subjects [5]. For one thing, the use of infrared as a light source may have eliminated the possibility of reflex tearing compared with the relatively longer examination time needed for FD-OCT [15]. The scan time of the FD-OCT is 0.16 s [15]. However, it usually takes longer for an examiner to focus to obtain a clear image than the keratograph. Furthermore, compared with the sagittal view image provided by FD-OCT, the keratograph did not automatically delineate the eyelid margin or the upper margin of the lower meniscus in plain frontal keratograph images. It was indicated that optical distortion in FD-OCT is usually caused by converting an image from the optical space into the physical space in FD-OCT algorithms [15,18]. The differences in image processing and operating principles between the two devices should be taken into consideration when comparing the between-device results. Despite the discrepancy in FD-OCT with higher TMH values, the TMH measured with the keratograph also showed a close correlation with that measured with FD-OCT in both groups in the Spearman analysis.

Though several studies have assessed the agreement between the keratograph and FD-OCT in the measurement of the TMH, different opinions have been reached [5,15]. One study enrolled DED patients and indicated that the TMH measured with Keratograph 5M closely correlated with the TMH measured with FD-OCT and had good repeatability and reliability [15]. Specifically, the TMH measured with Keratograph 5M tended to be lower in higher TMHs [15]. Another study with normal subjects showed no significant differences in the TMH values obtained using each device and the Bland–Altman plot showed poor agreement between Keratograph 5M and FD-OCT [5]. For the measures concerning agreements, CoV was >0.24% and the ICCs were low, indicating poor correlation between the two methods. An explanation for these conflicting results may be the different inclusion criteria for the subjects. Thus, we enrolled both healthy subjects and DED patients and analyzed them separately. Subsequently, poor agreement between the two devices was shown in the two populations. First, FD-OCT demonstrated better repeatability in TMH measurements than the data from the keratograph in both groups, as evidenced by the higher ICC. The use of FD-OCT to measure the TMH has been demonstrated to offer low variability and good reproducibility and repeatability and has been described as a good diagnostic method with high sensitivity and specificity for DED [5,8,9,10,11,19]. The blurry boundary of the tear meniscus with the keratograph compromised the repeatability of its measurements, possibly limiting its clinical use. Moreover, the ICC value with the keratograph in the DED group was less than 0.8 and the ICC value in the control group was 0.817, while good repeatability was obtained with FD-OCT in both groups. In addition to the technical reasons mentioned above, the unstable tear film and uncomfortable symptoms in DED patients may account for the low repeatability.

Furthermore, we studied the relationship between noninvasive TMH measurement methods and their associated clinical assessment of TBUT and Schirmer values. In addition to the weak correlation between the keratograph TMH and Schirmer scores in the DED group, no significant results were found. Previous studies on the relationship between these noninvasive methods and their associated clinical assessments have provided inconsistent results [6,20].

This study has some limitations. The lack of intersession reproducibility and interoperator reproducibility in the current study may have prevented us from comprehensively analyzing the results. In addition, we did not consider clinical dry-eye grading in our analysis. Different levels of severity of DED in studies may lead to different results. Furthermore, the relatively small sample size may be another limitation. These limitations would need to be addressed in future studies.

In conclusion, FD-OCT offered good repeatability for TMH measurements in both groups, while the keratograph only showed good repeatability in healthy subjects. However, both devices showed good diagnostic efficacy in differentiating healthy patients from DED patients. 

## Figures and Tables

**Figure 1 jcm-11-01343-f001:**
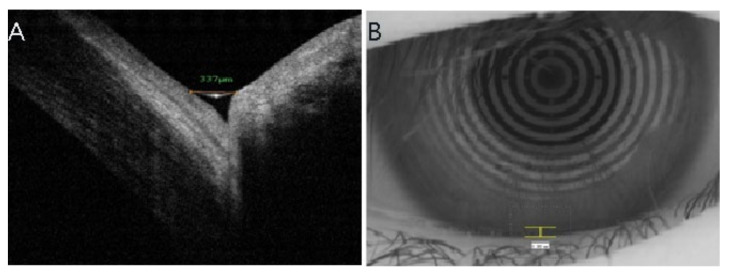
FD-OCT vertical line scan cross-sectional image of the lower tear meniscus showing the TMH. The TMH was measured using FD-OCT software (**A**). The measurement was performed over the 6 o’clock position of the cornea. The TMH was measured using Keratograph software (**B**). TMH, tear meniscus height; FD-OCT, Fourier-domain optical coherence tomography.

**Figure 2 jcm-11-01343-f002:**
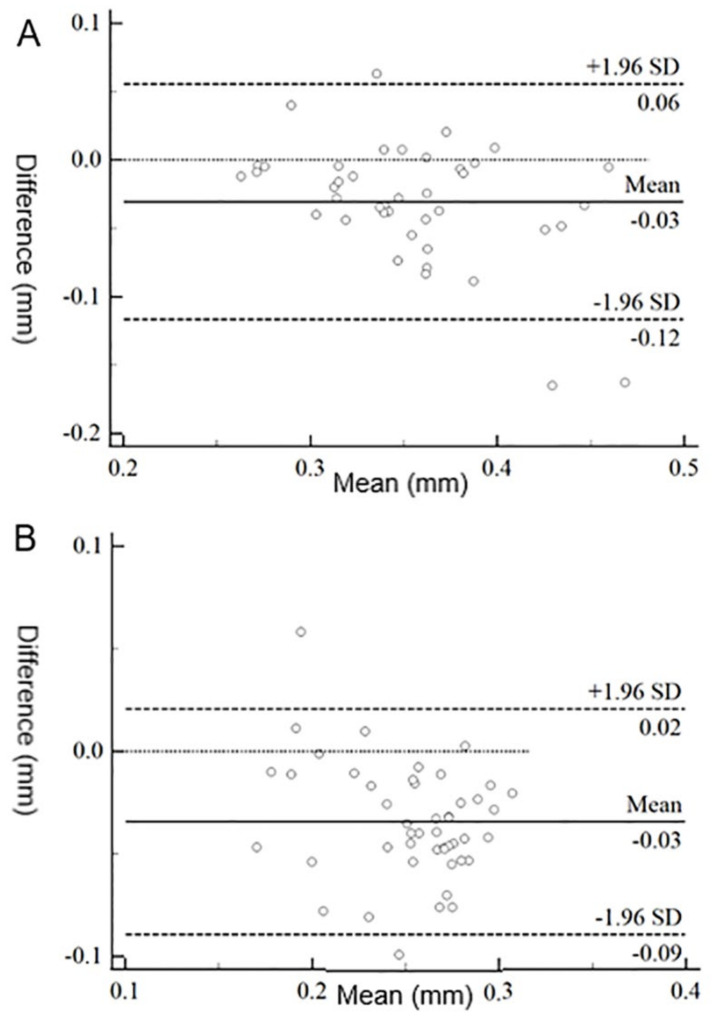
Bland-Altman plot of the TMH measurements made with keratograph and FD-OCT in the control group (**A**) and DED group (**B**). DED, dry eye disease; THM, tear meniscus height; FD-OCT, Fourier-domain optical coherence tomography.

**Figure 3 jcm-11-01343-f003:**
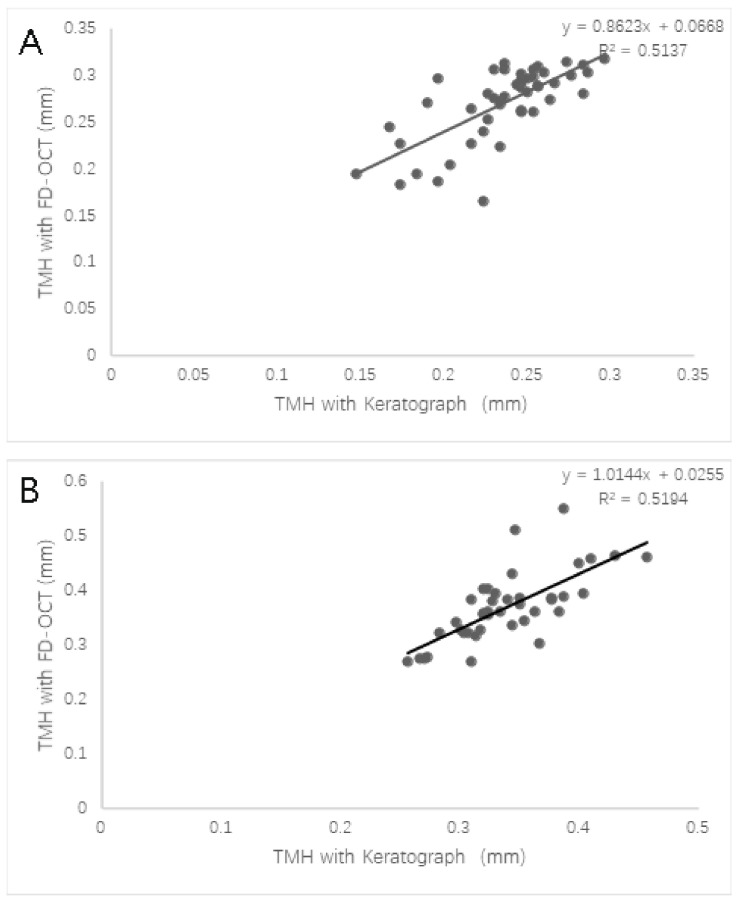
Correlation between the TMH measured with the keratograph and with FD-OCT in the DED group (**A**) and control group (**B**). DED, dry eye disease; TMH, tear meniscus height; FD-OCT, Fourier-domain optical coherence tomography.

**Figure 4 jcm-11-01343-f004:**
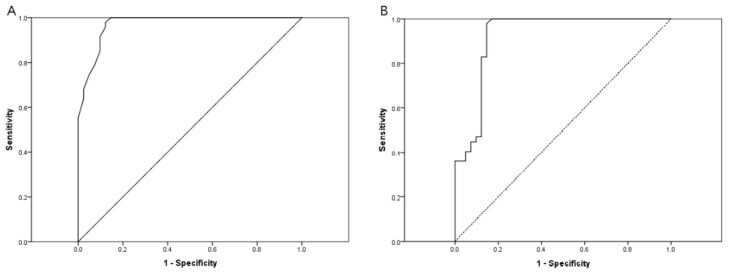
Comparison of the ROC curve for TMH measurements using keratograph and FD-OCT between the control group and DED patients. The AUC of the TMH measurements using the keratograph and FD-OCT were 0.971 (95% confidence interval (CI), 0.942–1.000; *p* < 0.001) (**A**) and 0.923 (95% CI, 0.861–0.986; *p* < 0.001), respectively (**B**). DED, dry eye disease; TMH, tear meniscus height; FD-OCT, Fourier-domain optical coherence tomography.

**Table 1 jcm-11-01343-t001:** Grading criteria for ocular surface discomfort.

Ocular Discomfort Symptoms	Scale from 0 to 4
1. Eyes that are sensitive to light?	0: none
2. Eyes that feel gritty?	1: occasionally
3. Painful or sore eyes?	2: half time
4. Blurred vision?	3: often
5. Poor vision?	4: always

**Table 2 jcm-11-01343-t002:** Patient and ocular characteristics in the DED and control groups.

	DED (*n* = 47)	Normal (*n* = 41)	*p*
Male (%)	53.19%	46.34%	0.521
Age (y)	46.74 ± 15.05	26.88 ± 6.73	<0.001
TMH with keratograph (mean ± SD (mm))	0.24 ± 0.03	0.34 ± 0.05	<0.001
TMH with OCT (mean ± SD (mm))	0.27 ± 0.04	0.37 ± 0.06	<0.001
TBUT (s)	3.51 ± 0.93	8.44 ± 1.42	<0.001
Schirmer I (mm)	6.19 ± 2.98	23.37 ± 6.50	<0.001

DED, dry eye disease; y, years; THM, tear meniscus height; OCT, optical coherence tomography; TBUT, tear film breakup time; s, seconds.

**Table 3 jcm-11-01343-t003:** Intraoperator repeatability of TMH measurements.

		Mean ± SD (mm)	Sw (mm)	2.77 Sw	CoV (%)	ICC (95% CI)
DED group	Keratograph	0.24 ± 0.03	0.02	0.05	7.02	0.789 (0.644–0.878)
	OCT	0.27 ± 0.04	0.02	0.04	5.89	0.859 (0.644–0.935)
Control group	Keratograph	0.34 ± 0.05	0.02	0.06	6.10	0.817 (0.696–0.895)
	OCT	0.37 ± 0.06	0.03	0.07	7.14	0.845 (0.753–0.909)

DED, dry eye disease; THM, tear meniscus height; OCT, optical coherence tomography; CoV, coefficient of variation; ICC, intraclass correlation coefficient; Sw, within-subject SD.

**Table 4 jcm-11-01343-t004:** Correlations between TMH measurements and clinical test results in two devices.

	DED Group				Control Group			
	TBUT		Schirmer		TBUT		Schirmer	
	*p*	r	*p*	r	*p*	r	*p*	r
TMH with keratograph	0.529	0.094	0.001	0.460	0.493	0.110	0.791	−0.043
TMH with FD-OCT	0.724	−0.053	0.154	0.211	0.840	0.033	0.982	−0.004

DED, dry eye disease; THM, tear meniscus height; OCT, optical coherence tomography; TBUT, tear break-up time; Spearman correlation test *p* < 0.05; r = correlation coefficient.

## Data Availability

The data presented in this study are available on request from the corresponding author.

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
