# Peer review of "Application of Keratograph and Fourier-Domain Optical Coherence Tomography in Measurements of Tear Meniscus Height"

_jcm, 2022, doi:10.3390/jcm11051343_

Round 1

Reviewer 1 Report

The authors compared the tear meniscus height in dry eye patients and normal subjects using the Keratograph and FD-OCT. It is a clinically meaningful study, but there are some parts that need to be corrected in this manuscript.

1.    Authors are needed to use the generic name of keratograph. In the manuscript, “Keratograph,” “keratography,” “Oculus Keratograph,” or “Keratograph 5M” are used in a mixed way. 
2.    In  introduction, authors stated that "the TMH was measured three times with each device by two separate examiners.” However, in method, “Each instrument was operated and measured by a single examiner who was masked to the study.” How many examiners participated in this study? 
3.    When a device first appears in Introduction, the device name, company name, and country name should appear.
4.    Please add Keratograph version number when presenting the results of previous studies.
5.    “Our findings indicate worse intraoperator repeatability for the keratograph than for the FD-OCT measurements for both groups. Comparable repeatability in TMH was observed with FD-OCT in both the DED and control groups, whereas better reproducibility for the keratograph measurements was seen in the control group than in the DED group.” 
This paragraph should be corrected because the authors did not measure reproducibility in this study. 
6.    In figure legends, please use “Keratograph” instead of “Keratograph 5M.” 
7.    In Table 3, please reduce the number of significant digits from four to three.
8.    In discussion, “For one thing, the use of infrared as a light source may have eliminated the possibility of reflex tearing compared with the relatively longer examination time needed for FD-OCT [15].” Image acquisition time of FD-OCT is very short. Authors should add more appropriate reasons or provide image acquisition time to support this sentence. 
9.    In discussion, “The TMH measured with the keratograph tended to be lower than the TMH measured with FD-OCT, which is in line with previous studies [5,15]” In previous studies, how much lower was the TMH measured by the Keratograph than the FD-OCT?
10.    Please provide the previous study results regarding the agreements in detail. 

Author Response

The authors compared the tear meniscus height in dry eye patients and normal subjects using the Keratograph and FD-OCT. It is a clinically meaningful study, but there are some parts that need to be corrected in this manuscript.

  1. Authors are needed to use the generic name of keratograph. In the manuscript, “Keratograph,” “keratography,” “Oculus Keratograph,” or “Keratograph 5M” are used in a mixed way.

Response: We did in the revised paper.

  1. In introduction, authors stated that "the TMH was measured three times with each device by two separate examiners.” However, in method, “Each instrument was operated and measured by a single examiner who was masked to the study.” How many examiners participated in this study?

Response: Each device was operated by one single person. Thus, two examiners participated in this study. We changed the expression in the abstract part.

  1. When a device first appears in Introduction, the device name, company name, and country name should appear.

Response: We did in the revised paper.

  1. Please add Keratograph version number when presenting the results of previous studies.

Response: We did in the revised paper.

  1. “Our findings indicate worse intraoperator repeatability for the keratograph than for the FD-OCT measurements for both groups. Comparable repeatability in TMH was observed with FD-OCT in both the DED and control groups, whereas better reproducibility for the keratograph measurements was seen in the control group than in the DED group.” This paragraph should be corrected because the authors did not measure reproducibility in this study.

Response: TMH was repeated for three times with each device and each device was operated by one single operator. Thus we checked the intraoperator repeatability with Sw, 2.77 Sw, CoV and ICCs. The lack of intersession reproducibility and interoperator reproducibility in the current study were a big limitation. We changed the word “reproducibility” to “repeatability” in the revised paper.

  1. In figure legends, please use “Keratograph” instead of “Keratograph 5M.”

Response: We did in the revised paper.

  1. In Table 3, please reduce the number of significant digits from four to three.

Response: We did in the revised paper.

  1. In discussion, “For one thing, the use of infrared as a light source may have eliminated the possibility of reflex tearing compared with the relatively longer examination time needed for FD-OCT [15].” Image acquisition time of FD-OCT is very short. Authors should add more appropriate reasons or provide image acquisition time to support this sentence.

Response: Image acquisition time of FD-OCT was about 0.16 seconds, but it usually took longer time for an examiner to focus to obtain a clear image than the keratograph. We added the expression in the revised paper.

  1. In discussion, “The TMH measured with the keratograph tended to be lower than the TMH measured with FD-OCT, which is in line with previous studies [5,15]” In previous studies, how much lower was the TMH measured by the Keratograph than the FD-OCT?

Response: In published data, the TMH measured with the keratograph was 0.07mm lower than the TMH measured with FD-OCT in DED patients [15], wheras the difference was 0.01mm in healthy subjects [5].

We added it in the revised paper.

  1. Please provide the previous study results regarding the agreements in detail.

Response: We added the previous study results regarding the agreements between the two devices in detail in the forth paragraph in the revised paper.

Reviewer 2 Report

The Authors present a paper evaluating two techniques in the field of dry eye disease, and their continued efforts on such an important topic should be lauded; a minor revison is suggested before possible publication.
In the Materials and Methods section (reference 7) more information about the questionnaire should be acknowledged and reported in the manuscript.

Author Response

The Authors present a paper evaluating two techniques in the field of dry eye disease, and their continued efforts on such an important topic should be lauded; a minor revison is suggested before possible publication.

In the Materials and Methods section (reference 7) more information about the questionnaire should be acknowledged and reported in the manuscript.

Response: We added the questionnaire as table 1 in the revised paper.

Reviewer 3 Report

This clinical study investigated the intraoperator repeatability of tear meniscus height (TMH) measured with keratography and Fourier domain optical coherence tomography (FD-OCT), and agreement between the methods was assessed. A total of 47 eyes with dry eye disease (DED) and 41 healthy eyes were analyzed. The authors found a close correlation between TMH measured with keratography and FD-OCT in both groups, and a poor agreement between devices in both groups using the Bland Altman plot. The authors conclude that both devices have excellent diagnostic accuracy in differentiating normal patients from DED patients. FD-OCT TMH measurements were more reliable than keratography data in DED group.

Author Response

This clinical study investigated the intraoperator repeatability of tear meniscus height (TMH) measured with keratography and Fourier domain optical coherence tomography (FD-OCT), and agreement between the methods was assessed. A total of 47 eyes with dry eye disease (DED) and 41 healthy eyes were analyzed. The authors found a close correlation between TMH measured with keratography and FD-OCT in both groups, and a poor agreement between devices in both groups using the Bland Altman plot. The authors conclude that both devices have excellent diagnostic accuracy in differentiating normal patients from DED patients. FD-OCT TMH measurements were more reliable than keratography data in DED group.

Response: Thank you for your comments. In this paper, we compared the TMH values obtained with keratography and FD-OCT in several methods, including the t test, repeatability test of Sw, 2.77 Sw, CoV and ICCs, Receiver operating characteristic (ROC) curves, Bland–Altman analysis  and Spearman correlation. Though some different conclusions were drawn, we carefully proofread the manuscript to correct logical errors to make it more fluently.

Round 2

Reviewer 1 Report

This study evaluated the repeatabilty, agreement, and accuracy of the TMH measurements between the Keratograph 5M and FD-OCT devices. The authors corrected the manuscript as recommended, however, more corrections are needed. 

  1. In the revised manuscript, the authors remove sentences regarding agreements between the two devices in whole manuscript. In abstract, there is no result regarding agreement as well.
  2. When a device first appears in Introduction, the device name, company name, and country name should appear.    

Author Response

Dear reviewer:

We really appreciate for the reviewers’ comments on our manuscript titled “Application of Keratograph and Fourier Domain Optical Co-herence Tomography in Measurements of Tear Meniscus Height”. In this article, we have revised the manuscript in accordance with the reviewers’ comments carefully. The revised texts are in red. In addition, our responses to the reviewers’ comments one by one are listed below. We hope the revised manuscript will be more acceptable for publication in your journal.

Best regards

Yours Sincerely,

Jiaxu Hong

Jiaxu Hong MD, PhD, MPA

Email: jiaxu_hong@163.com

Address: Department of Ophthalmology and Visual Science, Eye, and ENT Hospital, Shanghai Medical College, Fudan University, 83 Fenyang Road, Shanghai, China; Department of Ophthalmology, The Affiliated Hospital of Guizhou Medical University, Guiyang, China, Telephone: +86-021-64377134 Fax: +86-021-64377151

#Reviewer 1

This study evaluated the repeatabilty, agreement, and accuracy of the TMH measurements between the Keratograph 5M and FD-OCT devices. The authors corrected the manuscript as recommended, however, more corrections are needed.

1、In the revised manuscript, the authors remove sentences regarding agreements between the two devices in whole manuscript. In abstract, there is no result regarding agreement as well.

Response: We added the sentences regarding agreements between the two devices in the revised paper.

2、When a device first appears in Introduction, the device name, company name, and country name should appear.

Response: We added the device name, company name, and country name of the Keratograph when first appeared in the introduction in the revised paper. But many companies produce OCT, we didn’t add the device name, company name, and country name of OCT in its first appearance as previous study did (Arriola-Villalobos, P.; Fernández-Vigo, J.I.; Díaz-Valle. D.; et al. Assessment of lower tear meniscus measurements obtained with Keratograph and agreement with Fourier-domain optical-coherence tomography. Br. J. Ophthalmol. 2015, 99(8), 1120–1125).